**1**    **Quantification of hydraulic trait control on plant hydrodynamics and risk of hydraulic**

**2**    **failure within a demographic structured vegetation model in a tropical forest (FATES-**

**3**    **HYDRO V1.0)**

Chonggang Xu[1], Bradley Christoffersen[2], Zachary Robbins[1], Ryan Knox[3], Rosie A. Fisher[4],
Rutuja Chitra-Tarak[1], Martijn Slot[5], Kurt Solander[1], Lara Kueppers[3,6], Charles Koven[3], Nate
McDowell[7,8]
1: Earth and Environmental Sciences Division, Los Alamos National Laboratory, Los Alamos
NM, USA
2: School of Integrative Biological and Chemical Sciences, University of Texas Rio Grande
Valley, TX, USA
3: Lawrence Berkeley National Laboratory, Berkeley, CA USA
4. CICERO Centre for International Climate Research, Oslo, Norway
5: Smithsonian Tropical Research Institute, Apartado 0843-03092, Balboa, Ancon, Republic of
Panama
6: Energy and Resources Group, University of California, Berkeley, CA USA
7: Atmospheric Sciences and Global Change Division, Pacific Northwest National Laboratory,
Richland, WA, USA
8: School of Biological Sciences, Washington State University, Pullman, WA, USA

23    Corresponding author: Chonggang Xu (cxu@lanl.gov)

**Abstract**: Vegetation plays a key role in the global carbon cycle and thus is an important component within Earth system models (ESMs) that project future climate. Many ESMs are adopting methods to resolve plant size and ecosystem disturbance history using vegetation demographic models. These models make it feasible to conduct more realistic simulation of processes that control vegetation dynamics. Meanwhile, increasing understanding of the processes governing plant water use, and ecosystem responses to drought in particular, has led to the adoption of dynamic plant water transport (i.e., hydrodynamic) schemes within ESMs. However, the impact of plant hydraulic trait variation in trait-diverse tropical forests is understudied. In this study, we report on a sensitivity analysis of an existing hydrodynamics (HYDRO) model that is updated and incorporated into the Functionally Assembled Terrestrial Ecosystem simulator (FATES). The size and canopy structured representation within FATES is able to simulate how plant size and hydraulic traits affect vegetation dynamics and carbon/water fluxes. To better understand this new model system and its functionality in tropical forest systems in particular, we conducted a global parameter sensitivity analysis at Barro Colorado Island, Panama. We assembled 942 observations of plant hydraulic traits on 306 tropical plant species for stomata, leaves, stems, and roots, and determined the best-fit statistical distribution for each trait, which was used in model parameter sampling to assess the parametric sensitivity. We showed that, for simulated leaf water potential and loss of hydraulic conductivity across different plant organs, the four most important traits were associated with xylem conduit taper (buffers increasing hydraulic resistance with tree height), stomatal sensitivity to leaf water potential, maximum stem hydraulic conductivity, and the partitioning of total hydraulic resistance above vs. belowground. Our analysis of individual ensemble members revealed that trees at a high risk of hydraulic failure and potential tree mortality generally have higher conduit taper, maximum xylem conductivity, stomatal sensitivity to leaf

water potential, and lower resistance to xylem embolism for stem and transporting roots. We
expect that our results will provide guidance on future modeling studies using plant hydrodynamic
models to predict the forest responses to droughts, and future field campaigns that aim to better
parameterize plant hydrodynamic models.


## 1. Introduction

Tropical forests play a critical role in regulating regional and global climates (Bonan, 2008). Under ongoing and future climate change, they are subjected to substantial risks of climate extremes such as drought and heat waves (Mcdowell et al., 2018). Studies have already shown that tropical forests were experiencing elevated tree mortality rates due to mega droughts related to ENSO events. For example, the 2015–16 El Niño led to the death of an estimated $2.5 \pm 0.3$ billion stems in the Lower Tapajós river basin of the Amazon and the associated carbon loss had not yet been compensated by new plant growth three years after the event (Berenguer et al., 2021). Such extreme climate events are projected to increase in frequency and intensity under a warming future (Seneviratne et al., 2021). A statistical analysis based on the projection of 13 ESMs under a high greenhouse emission scenario showed that the frequency of extreme droughts as defined by rhizosphere soil moisture (occurring once every 50 years) could increase by a factor of nearly four and this increase would have a disproportionate impact on tropical forests (Xu et al., 2019). The high species diversity found in tropical forests may result in increased resilience to climate extremes, based on the demonstrated resilience of temperate forests in relationship to trait diversity (Anderegg et al., 2018). However, due to limited data to parameterize and constrain models for tropical forests, there is a large uncertainty in our predictive understanding of how tropical forests will respond to these climate extremes (Bonal et al., 2016). This tropical forest uncertainty is a key source of the global uncertainty in projections of land carbon fluxes and future climates (Arora et al., 2020).

Earth System Models (ESMs) have been developed to project future changes to the coupled
climate and biosphere system. Typically, 'big leaf' approximations of vegetation with no
explicit presentation of tree size and canopy structure have been used to predict the impact of
vegetation on carbon and water cycles. These models do not represent the fundamental
elements of vegetation dynamics including growth, mortality, competition, and their response
to disturbances. In the last decade, many ESMs have incorporated vegetation demographic
models (VDMs) that represent plant size, canopy structure and disturbance histories, with the
goal of better representing the competitive dynamics among different size classes of trees and
plant functional types in response to climate and disturbances (Fisher et al., 2018). Most of
these VDMs can differentiate plants' light, water and carbon use strategies and can thus
represent some part of the functional diversity of tropical forests (Massoud et al., 2019; Koven
et al., 2020).
Following the 'big leaf' model, water limitation on plant gas exchange in these VDMs is
generally calculated based on three factors: 1) soil water potential; 2) root distribution; and 3)
water potential for stomata openness and closure, all of which differ by plant functional types
(Koven et al., 2020). While these soil-moisture-dependent water limitation functions are able
to capture trait diversity in leaf-level stomatal behaviors, they fail to capture plant functional
diversity in many other observable plant hydraulic traits, such as xylem capacitance, water
potentials for loss of xylem hydraulic conductivity, stem hydraulic safety margin, and turgor
loss point (Hochberg et al., 2018). Many studies have shown that plant hydraulic traits play an
important role in plant responses to droughts (Su et al., 2022; Anderegg et al., 2016), which
could shape the landscape distribution of plant functional types (Kunert et al., 2021). In view
of this limitation, plant hydrodynamic models have been developed with the aim of better
simulating forest response to droughts (Powell et al., 2018; Christoffersen et al., 2016; Xu et
al., 2016; Kennedy et al., 2019; Mcdowell et al., 2013). These models not only incorporate
hydraulic functional diversity, but also mechanistically simulate the risk of plant mortality due
to hydraulic functional failure, as a result of  an inability to move water in the xylem due to
embolism in conduits (Hammond et al., 2019).

One key challenge for these plant hydrodynamic models is that they have many more

parameters than simple water limitation functions based on soil water potentials and thus
inherently possess more uncertainty in the model parameterization and subsequent simulations.
In this study, we describe the implementation of a hydrodynamic scheme within DOE-
sponsored functionally assembled terrestrial ecosystem simulator (FATES) (Koven et al.
2020), and assess this new configuration with two goals: 1) quantify the parametric sensitivity
of different hydraulic traits in determining plant hydrodynamics; and 2) identify key hydraulic
traits that are important for predicting the risk of mortality due to hydraulic failure. We expect
that our results will provide guidance on model parameterization for future modeling studies
using plant hydrodynamic models to predict tropical forest response to droughts, and future
field campaigns that aim to collect observational data that can be used to better parameterize
and benchmark plant hydrodynamic models.
**2. Methodology**
**2.1. Model description**

We use FATES, a VDM that is coupled within the Energy Exascale Earth System Model

(E3SM) (Caldwell et al., 2019). FATES represents size-structured groups of plants (cohorts)
and successional trajectory-based patches using the ecosystem demography approach (Fisher
et al., 2015; Moorcroft et al., 2001). FATES simulates growth by integrating photosynthesis
across different leaf layers for each cohort. FATES allocates this photosynthate to different
tissues including leaves, fine and coarse roots, and stem, based on the allometry of different
plant functional types, as well as a carbon storage pool (Fisher et al., 2015). Mortality within
FATES is simulated by several mechanisms, including carbon starvation caused by depletion
of the storage pool, hydraulic function failure, as well as impact mortality during disturbance,
fire, logging, freezing, age-related and 'background' constant turnover (Fisher et al., 2015;
Huang et al., 2020; Fisher et al., 2010; Needham et al., 2020).
### 2.1.1.  Plant Hydrodynamics
The default (non-hydrodynamic) FATES model contains a simplistic algorithm that
approximates plant hydraulic failure thresholds based on soil water potential.  An important
feature of the plant hydrodynamic scheme (HYDRO), which explicitly simulates water flow
from the soil through leaves to the atmosphere, is that it enables direct representation of percent
loss of conductance as a predictor of hydraulic failure mortality rates. FATES-HYDRO is
based on the hydrodynamic model implemented in the Traits-based Forest Simulator (TFS)
(Christoffersen et al., 2016) and the features most relevant to the present analysis are
summarized below. The model approximates water transport in a single vertical dimension,
approximating the canopy as a single leaf layer at the top of a beam, according to the Shinozaki
pipe model (Shinozaki et al., 1964) in which the hydraulic path length from the trunk base to
each leaf is assumed constant. Following the 'porous media' approach, the model simulates
the water transport across four main organs (leaves, stem- trunk/branches, transporting roots,
and absorbing roots) and different rhizosphere shells (Fig. 1).  Resistors connect the different
compartments.

143  The water flow is calculated based on water pressure gradients across different

144  compartments (rhizosphere, absorbing roots, transporting roots, stem, and leaf). Specifically,

145  flow between compartment $i$ and $i + 1$ ($Q_i$) is given by,

$$Q_i = -K_i \Delta h_i, \tag{1}$$

147  where $K_i$ is the total conductance (kg MPa$^{-1}$ s$^{-1}$) at the boundary of compartments $i$ and $i + 1$

148  and $\Delta h_i$ is the total matric potential difference between the compartments,

$$\Delta h_i = \rho_w g(z_i - z_{i+1}) + (\psi_i - \psi_{i+1}), \tag{2}$$

150  where $z_i$ is compartment elevation difference above (+) or below (-) the soil surface (m), $\rho_w$

151  is the density of water ($10^3$ kg m$^{-3}$), $g$ is acceleration due to gravity (9.8 m s$^{-2}$), and $\psi_i$ is

152  tissue or soil matric water potential (MPa). $K_i$ is treated here as the product of a maximum

153  boundary conductance between compartments $i$ and $i + 1$ ($K_{max,i}$), and the fractional

154  maximum hydraulic conductance of the upstream compartments ($FMC_i$ or $FMC_{i+1}$), which is

155  a function of the tissue water potential as follows,

$$FMC_i = \left[1 + \left(\frac{\psi_i}{P_{50,x}}\right)^{a_x}\right]^{-1}, \tag{3}$$

157  where $\psi_i$ is the compartmental water potential, $P_{50,x}$ is the water potential at 50% loss of

158  maximum conductivity for different plant tissues ( absorbing root, transporting root, stem), $a_x$

159  is the corresponding vulnerability curve shape parameter, with a larger number indicating a

160  steeper reduction of conductivity in response to more negative water potentials (Choat et al.,

161  2012). The maximum percentage loss of conductivity (PLC) across different organs [i.e., PLC$_i$

162  =100 (1-FMC$_i$) ] is used to measure the risk of tree mortality ($M_{hf}$) resulting from hydraulic

163  failure as follows,

$$M_{hf} = M_{hf,base} \frac{\max(0,\ PLC_{max,organ} - PLC_c)}{100 - PLC_c}, \tag{4}$$

where $PLC_c$ is the critical percentage loss of conductivity with risk of mortality, $PLC_{max,organ}$
is the maximum percentage loss of conductivity across different organs, $M_{hf,base}$ is the
baseline mortality rate [fraction/year] when percentage loss of conductivity exceeds $PLC_c$. In
this version of model, we assume that xylem cavitation can fully recover as long as the trees
do not die.
The previous version of this model (TFS-Hydro) presented water in terms of relative water
content (RWC; g $H_2O$ g$^{-1}$ $H_2O$ at saturation) in line with most empirical work on plant water
relations. While the underlying equations remain unchanged, here we present water in terms
of volumetric water content ($\theta$; m$^3$ $H_2O$ m$^{-3}$ plant tissue), since this what is accounted by the
model and is consistent with what is tracked in the soil as well. The two quantities are related
via the equation RWC $= \theta/\theta_{sat}$, where $\theta_{sat}$ indicates saturated volumetric water content. The
water potential for tissue $x$ [$\psi_x$] is related to $\theta_x$ (the PV curve) following three stages of water
tissue drainage as follows (Tyree and Yang, 1990; Bartlett et al., 2012),

$$\psi_x = \begin{cases} \psi_{0,x} + m_{cap}\left(\frac{\theta_x}{\theta_{sat,x}} - 1\right) & \theta_{ft} < \theta_x \leq \theta_{sat,x} \\ \psi_{sol}(\theta_x) + \psi_p(\theta_x) & \theta_{tlp,x} < \theta_x \leq \theta_{ft,x} \\ \psi_{sol}(\theta_x) & \theta_{r,x} < \theta_x \leq \theta_{tlp,x} \end{cases} \quad (5)$$


Stage one applies to stem and roots only and represents the water draw from capillary reserves
(embolized conduits or airspaces in wood) when wood water content is in between full turgor
($\theta_{ft} = RWC_{ft}\,\theta_{sat,x}$) and saturation ($\theta_{sat,x}$) and only represents a small fraction of the total
PV curve. It is linear with constant slope $m_{cap}$ = 11.3 MPa m$^3$ m$^{-3}$ and $RWC_{ft}$ = 0.958 as
estimated from sapwood PV curves on 28 tropical and subtropical species (Christoffersen et
al. 2016). $RWC_{ft}$ is assumed to be 1.0 in leaves. Xylem water potential is assumed zero at full
saturation. The second stage is between full turgor ($\theta_{ft,x}$) and the turgor loss point ($\theta_{tlp,x}$),
when the xylem water potential is in balance with solute ($\psi_{sol}[\theta_x]$) and pressure water
potential ($\psi_p[\theta_x]$) of living cells. The third stage is after the turgor loss point ($\theta_{tlp,x}$), but
above the point of residual water content ($\theta_{r,x} = RWC_{r,x}\,\theta_{sat,x}$) where the water potential is
only a function of the solute water potential. $RWC_{r,x}$ is synonymous with the apoplastic
fraction (Bartlett et al. 2012).

The solute water potential is given as,

$$\psi_{sol}[\theta_x] = \frac{\pi_0(\theta_{sat,x}RWC_{ft} - \theta_{r,x})}{(\theta_x - \theta_{r,x})}, \tag{6}$$

where $\pi_0$ is the tissue osmotic potential at full turgor. The pressure potential is calculated as
follows,
$$\psi_p[\theta_x] = -|\pi_0| + \varepsilon\frac{(\theta_x - \theta_{sat,x}RWC_{ft})}{(\theta_{sat,x}RWC_{ft} - \theta_{r,x})}, \tag{7}$$

where $\varepsilon$ is the bulk elastic modulus (MPa).

The realized conductivity of the above ground portion of the plant per unit of leaf area (

$K_{l,max,tree,ag}$) is calculated based on xylem hydraulic conductivity at petiole ($k_{s,max,petiole}$),
aboveground tree height (H, meters), and a xylem taper factor ($X_{tap}$) as follows,
$$K_{l,max,tree,ag} = \frac{k_{s,max,petiole}}{H(\frac{A_l}{A_s})}X_{tap}, \tag{8}$$

where $k_{s,max,petiole}$ is scaled from the xylem conductivity measured from the branch ($k_{s,max}$)
(Christoffersen et al., 2016). $\frac{A_l}{A_s}$ [i.e., la2sa in Table 1] is the ratio of leaf area ($A_l$) to sapwood
area ($A_s$). $X_{tap}$ is the xylem taper factor representing the ratio of aboveground xylem
conductance with taper to that without, which for intermediate values of conduit taper (p_taper
= 1/6; see below) represents a factor increase in total conductance of 23–50 for trees of heights
10–30 meters  (Christoffersen et al., 2016).  Savage et al. (2010) highlighted how opposing
selective forces will both increase hydraulic conductance by the tapering of conduit radii
($p\_taper > 0$) while at the same time protect against embolism by minimizing conduit taper
(no taper implies $p\_taper = 0$). They defined $p\_taper$ as the exponent on an external branching
parameter (2 daughter branches per parent branch in their model) that sets the degree of internal
branching of xylem conduits (and thus the tapering of conduit radii as well) and, using a fractal
network model, derived an effective exponent $q$ that describes how aboveground conductance
increases with tree size. $q$ is a monotonically increasing and saturating function of the taper
exponent $p$ (see Fig 2b of Savage et al. 2010); we used this relationship to estimate $q$, and thus
$X_{tap}$ in eq. (8) as
$$X_{tap} = \left[\frac{r_{base}}{r_{petiole}}\right]^{q_{tap}-q_{notap}} , \qquad (9)$$
where $r_{base}$ and $r_{petiole}$ are the trunk and petiole radii, respectively. The ratio $r_{base}/r_{petiole}$ is
related to tree height following the fractal tree model of Savage et al. (2010) (see equations
S12-S13 in Christoffersen et al. 2016).
Eq. (8) only gives the aboveground component of whole-plant conductance. In the
absence of a simple first-principles approach to estimating the belowground component, we
estimate the total tree maximum conductance (above- and belowground components) as
$$K_{max,tree,total} = R_{frac,stem}K_{max,tree,ag} , \qquad (10)$$
where $R_{frac,stem}$ is the fraction of total resistance that is aboveground.
Stomatal conductance [$g_s$, $\mu$mol m$^{-2}$ s$^{-1}$] is simulated through a modified Ball-Berry
equation,
$$g_s = g_0 + g_1 \frac{A_n}{C_s/P_{atm}} h_s , \qquad (11)$$
where g₁ is the stomatal conductance slope in response to environmental condition changes,
$g_0$ is the minimum (cuticular) stomatal conductance ($\mu$mol m$^{-2}$ s$^{-1}$), $C_s$ is the leaf surface $CO_2$
partial pressure (Pa), $P_{atm}$ is the atmospheric pressure (Pa), $h_s$ is the leaf surface humidity,
and $A_n$ is leaf net photosynthesis rate ($\mu$mol $CO_2$ m$^{-2}$ s$^{-1}$). Stomatal conductance (i.e.,
both $g_0$ and $g_1$) is further modified by a plant water stress factor, $\beta$, calculated as

$$\beta = [1 - \left(\frac{\psi_{leaf}}{P_{50,gs}}\right)^{a_{gs}}]^{-1} , \tag{12}$$

where $\psi_{leaf}$ is the leaf water potential, $P_{50,gs}$ is leaf water potential at 50% loss of maximum
stomatal conductance, and $a_{gs}$ is the stomatal vulnerability shape parameter.

The total fine root surface area affects the amount of water a plant can take up through its

influence on rhizosphere conductance and is determined by both the specific root length (srl)
and absorbing root radius (rs2). Specifically, the model has a specified number of soil shells
(5 in this study) around fine root surfaces and the conductance between soil shell k+1 and k ,
$K_{shell,k}$, is calculated as,

$$K_{shell,k} = K_s \frac{\pi \, l_{aroot,common}}{\ln(r_{k+1}/r_k)} , \tag{13}$$

where $r_k$ is the mean radi of kth shell, $l_{aroot,common}$ is the total length of absorbing roots
calculated as a product of total fine root biomass and specific root length (srl). $K_s$ is set to be
the conductance for soil ($K_{soil}$) when k>1. For k = 1,

$$K_s = \frac{1}{\frac{1}{K_{soil}} + \frac{1}{K_{root\_soil}}} , \tag{14}$$

where $K_{root\_soil}$ is the conductance between fine root surface and soil. An update to the TFS-
Hydro approach is to make this conductance direction-specific, in view that water loss rate
from root could be substantially lower than water uptake rate either through osmatic regulation
(Dichio et al., 2006) or by lacunae caused by rupture of cortical cells (North and Nobel, 1992)
during drought. It is determined by either the maximum uptake of water per unit of absorbing
root surface area ($k_{r1,max}$, kg m$^{-1}$ s$^{-1}$ MPa$^{-1}$) when root water potential is more negative than
adjacent rhizosphere soil water potential, or the maximum root water loss rate per unit surface
area ($k_{r2,max}$, kg m$^{-1}$ s$^{-1}$ MPa$^{-1}$) when rhizosphere water potential becomes more negative than
root water potential, which may occur, for example, in frozen soils or in very dry soil layers
(Schmidhalter, 1997).

The plant hydrodynamic representation and numerical solver scheme within FATES-

HYDRO follows the 1-D solver laid out by Christoffersen et al. (2016), which is the default
solver in FATES-HYDRO and used in this study. The model also has an option of a 2-D solver,
which is slower and detailed by Fang et al. (2022) and Lambert et al. (2022). The equations
are solved for tissue water content at a 30 minutes time step. We made a few modifications to
accommodate multiple soil layers and improve the numerical stability. First, to accommodate
the multiple-soil layers, we sequentially solve the Richards' equation for each individual soil
layer, with each layer-specific solution proportional to each layer's contribution to the total
root-soil conductance. Second, to improve the numerical stability, we now linearly interpolate
the pressure/volume curve beyond the residual and saturated tissue water content to avoid the
rare cases of overshooting in the numerical scheme under very dry or wet conditions. See the
Supplementary Information [HYDRO_DESCRIPTION.pdf] for further details of the
implementation.
**2.1.2. Non-hydrodynamics processes**

FATES-HYDRO can be coupled to different host land models (HLMs) including the

E3SM land model (ELM) (Caldwell et al., 2019) or the Community Terrestrial Systems Model
(CTSM) (Lawrence et al., 2019). In this study, the model is coupled to ELM. In this section,
we layout the key non-hydrodynamic processes in the FATES or the ELM for a better
understanding of parameter importance in the results.

Canopy radiative transfer is calculated using a multi-layer scheme based on the iterative

Norman radiation scheme (Norman, 1979). Leaf and stem area is binned into a matrix of
canopy layer, leaf layer and plant functional types. Reflectance, absorption, and transmittance
are calculated for each leaf layer. Between canopy layers, light streams are averaged between
plant functional types (PFTs), such that all PFTs in understory layers receive equal radiation
on their top leaf layer.  Fractional absorption of visible and near infra-red light is calculated
separately for direct and diffuse light. For the direct stream, transmitted and reflected light is
converted into diffuse fluxes. In FATES, the absorbed PAR is used to calculate photosynthesis
rates for each of the canopy layer x leaf layer x PFT bins, after which rates across layers are
re-aggregated into cohort level carbon fluxes. Please see the Supplementary file in Fisher et al.
(2015) for details.

The energy balance is handled by the host land model. In this study, it is based on the land

component of DOE's Exascale Energy Earth System Model (E3SM). The E3SM land model
(ELM) is based on the Community Land Model 4.5 (Oleson, 2013). Specifically, in ELM, the
average canopy temperature is calculated based on the energy balance of latent heat, sensible
heat, and absorbed radiation as determined by the radiative transfer model. The latent heat is
determined by the transpiration, which is determined by the vapor pressure deficit from inside
of leaf to the air, canopy stomatal conductance, and boundary layer conductance. FATES
calculated mean canopy stomatal conductance averaged across different cohorts, which is fed
to ELM to calculate the energy balance. The Newton-Raphson numerical scheme is used to
solve for the canopy temperature.
All aspects of soil water balance (infiltration, water transfer among soil layers, and
drainage) happen at the 'column' scale at 30-min time steps and are handled within the Host
Land Model (see Oleson et al. 2013 for a detailed description of hydrology in CLM4.5, the
parent model of ELM). FATES-HYDRO handles soil water operations at the patch and cohort
scales. It simulates root water uptake and changes in plant water potential from roots to leaves
based on current time step transpiration. The belowground conductance for each soil layer is
weighted by root biomass with an exponential vertical distribution. Sections 2 and 3 in the
Supplement of this manuscript provide full details on boundary conditions, sequence of
operations among HYDRO and the HLM, downscaling of soil moisture to rhizosphere shells,
and downscaling of transpiration from the patch to individual scale.
**2.2. Sensitivity analysis**
We identified 35 parameters for the FATES-HYDRO model to conduct the parametric
sensitivity analysis (Table 1). To estimate the parameter distributions, we started with
published meta-analyses (Christoffersen et al., 2016; Choat et al., 2012; Bartlett et al., 2012;
Bartlett et al., 2014; Bartlett et al., 2016; Klein, 2014) and supplemented them with select new
data from individual studies. Focal data were tissue- or individual-level hydraulic traits
spanning water transport and embolism resistance, tissue water storage and retention (PV curve
traits), hydraulic architecture (i.e., leaf area to sapwood area ratio), stomatal responses to
dehydration, and fine root traits (Table 1). For each dataset, we standardized taxonomic names
using the TNRS package in R (Boyle et al., 2013). This allowed us to join datasets together
based on species, averaging multiple observations per species if necessary, resulting in a
species-specific sparse matrix of all hydraulic traits for all databases and individual studies that
we compiled. This pantropical hydraulic trait dataset is included in the Supporting Information
[traits_master_trop.csv].

This trait dataset consisted of anywhere from 1 - 323 observations for each trait, where
each observation corresponds to a different species (multiple observations for the same species
are first averaged; see above). Before fitting distributions to these data, some traits were first
transformed to be positive (e.g., P50) or normalized within [0, 1] when upper and lower bounds
were well-defined (Table 1). Then, for each trait separately, we used the fitdistr package in R
to estimate best-fit parameters for uniform, beta, normal, lognormal, and gamma statistical
distributions in order to estimate central tendencies and spread for each trait. The distribution
with the largest log likelihood and best-fit parameters are given in Table 1. Each model
simulation consisted of a single PFT: all trees (across all cohort sizes and patches) had the
same traits.
We augmented observations with extratropical data to increase sample size for traits with
less than three tropics-specific observations. When trait data observations were not present, we
used a uniform distribution bounded on our best estimate of the theoretical range (Table 1).
As there is limited data on roots, we used the same distribution as that for branches if data were
lacking. Because our goal is to understand the model behaviors as determined by different
hydraulic traits, we assumed independence among traits. As we focused on the hydraulic traits
in this study, we used non-hydraulic trait values based on an optimal set of parameters that best
fit observed water and carbon fluxes in a set of FATES simulations run without hydrodynamics
(Koven et al., 2020).
We used the Fourier Amplitude Sensitivity Test (FAST) to assess the relative importance
of parameters in determining the variance of model outputs (Xu and Gertner, 2011a). The main
idea of FAST is to assign periodic signals in the sampled parameter values and use Fourier
transformation to identify the signals in the outputs. Sampled parameter values are based on
Latin hypercube sampling from the fitted statistical distributions (see previous section for more
details). We ran 1000 ensemble simulations of the FATES-Hydro to derive model outputs of
water potential and fraction of maximum conductivity. For each ensemble simulation, each
plant hydraulic trait was assigned with a random draw from each trait's distribution, and the
samples for different traits are randomly combined to sample the observed plant hydraulic trait
space for sensitivity analysis.
We used the Uncertainty Analysis and Sensitivity Analysis (UASA) tool
(https://sites.google.com/site/xuchongang/uasatoolbox) to estimate the parametric sensitivity
index, which is calculated based on the ratio of the partial variance in the model output
attributed to a specific parameter to the total variables in the model output. For details, please
refer to Xu and Gertner (2011a). We ran the model with 1000 ensemble members, in view that
an order of 100 times effective important number of parameters, which we estimate to be ~10,
is needed to achieve reasonable precision (Xu and Gertner, 2011b).
**2.3. Study area**
In this study, we used Barro Colorado Island (BCI), Panama, as our test site to evaluate
model behavior. We chose BCI because it has moderately strong dry and wet seasons that
allow us to assess the hydrodynamics under different levels of water availability. Moreover,
extensive field campaigns in recent years have provided comprehensive data needed for model
parameterization, initialization and climate drivers. Finally, we also leverage prior FATES
studies of non-hydraulic parameters at BCI (Koven et al., 2020).

BCI has an annual mean temperature of 26.3°C and an annual mean precipitation of 2656

mm with a strong seasonal precipitation signal. The dry season lasts from January to April,
with a mean precipitation of 228mm, while the wet season lasts from May-December with a
mean precipitation of 2428mm (Paton, 2020). In this study, we used hourly in-situ climate data
from 2008-2016 to drive the model. To run the model to equilibrium (in terms of soil moisture
content) takes 5-6 years, thus we choose February of 2016 as the target for analysis of dry
season hydrodynamics and August of 2016 as the target for analysis of wet season
hydrodynamics.
**2.4.    Model setup**

In this study, as our focus is on the plant hydrodynamics, we used the static stand structure

mode of FATES that turns off the processes of competition, growth and mortality, to instead
hold the ecosystem structure constant. This reduced-complexity configuration (Fisher and
Koven, 2020) thus exercises only the primarily fast-timescale-processes of photosynthesis,
transpiration, water transport, and plant hydrodynamics (i.e., change in hydraulic conductivity,
water storage, and water potentials in plant tissues). By using static stand structure mode, as in
Chitra-Tarak et al. (2021), we isolate hydraulic trait controls on simulated hydrodynamics and
avoid confounding, and potentially biased, feedbacks from resulting changes in forest
structure.   Using static stand structure mode also means that we do not need to spin up
vegetation state, thus reducing the simulation time. The forest stand structure, consisting of
tree size and composition for each patch, is initialized based on forest inventory data collected
in 2015 (http://ctfs.si.edu/webatlas/datasets/bci/).  As the majority of species in BCI are
evergreen broad leaf trees, we ran the model with one PFT with different hydraulic traits (Table
1) to assess their impact on the hydrodynamically relevant outputs including water potentials
and fraction of maximum conductivity for different plant organs including absorbing root,
transporting root, stem, and leaves.
One key benefit of utilizing a hydrodynamic model is its ability to simulate the risk of
hydraulic failure by considering the loss of conductivity in various plant organs. As FATES
model was ran on the static stand mode, we did not specifically simulate the tree mortality
resulting from the hydraulic failure as shown in Eq. (4). Instead, we used the maximum of loss
of conductance across the continuum of plant nodes [i.e., $PLC_{max,organ}$ in eq. (4)] to assess
the hydraulic failure risk. If $PLC_{max,organ}$ reaches critical threshold $PLC_c$, which is set to
50% (Adams et al., 2017), trees are assumed to be faced with a high risk of mortality. Using
the ensemble simulations, we also aim to identify the most vulnerable plant organs and the
critical parameters that influence the likelihood of hydraulic failure. The HYDRO model only
considers the stem node (Fig. 1) without explicitly simulating the branch. In this analysis, we
calculated the branch vulnerability by using the PLC curve of xylem and the leaf water
potential, which approximates the water potential at the tip of the branch. The model does not
explicitly consider xylary or extraxylary resistance within and outside the leaf midrib.
FATES simulates the carbon and water fluxes for different size classes of trees. The forest
has 137 cohorts with diameters ranging from 10 cm to >2 meters and height ranging from 1 to
38 meters (see Fig. S1 for size distributions). Because large trees experience more fluctuations
in environmental conditions in the canopy and higher risk of mortality due to drought (Bennett
et al., 2015), we focused on hydrodynamic behaviors for large trees with diameter at breast
height (DBH) more than 60 cm; however, for comparison, we also derived the sensitivity for
smaller trees with DBH less than 60 cm.
**3. Results**
Our results showed that the simulated ranges across the ensemble of leaf water potential
(Fig. 2) and loss of conductivity (Fig. 3) are large. For leaf water potential of large trees with
diameter > 60 cm, the 95% percentile ranges are from -5 MPa to -0.5 MPa and -3 MPa to -0.5
MPa for February (dry) and August (wet) 2016, respectively.  Correspondingly, the fraction of
maximum stem hydraulic conductivity is much higher during August compared to February
(Fig. 3); however, in both months, the modeled range spans almost the full range of between 0
and 1.  For smaller trees with diameter less than 60 cm, our results show that smaller tree
experienced less negative water potential (Fig. S2 and Fig. 2) and lower loss of hydraulic
conductivity (Fig. S3 and Fig. 3).
Based on the FAST sensitivity indices (i.e., the variance in model output contributed by
different parameters), the key parameters that control the water potentials of different plant
organs (leaf, stem and root) for large trees (diameter >60 cm) include the taper exponent for
hydraulic conductivity (*p_taper*), the water potential leading to 50% loss of stomatal
conductance (*p50_gs*), maximum hydraulic conductivity for the stem (*kmax_node_stem*), and
the fraction of total hydraulic resistance in the above ground section (*rfrac_stem*), in decreasing
order (Fig. 4). For the fractional loss of conductivity, the most important parameter is the water
potential leading to 50% loss of hydraulic conductance ($P_{50}$) for the corresponding organs (Fig.
5). Other important parameters are similar to those for simulated water potentials. Notably, the
organ-specific $P_{50}$ values are more important for the dry month (February) compared to the
wet month (August). For the wet month of August, *p_taper* is the dominant parameter
controlling the pre-dawn and midday loss of hydraulic conductivity, while organ-specific $P_{50}$
parameters are the second most important. For smaller trees with diameter less than 60 cm, the
corresponding parametric sensitivity patterns are similar to those of larger trees (Fig. S4 and
Fig. S5); however, compared to larger trees, the parametric sensitivity of *p_taper* for simulated
leaf water potential becomes lower for smaller trees (Fig. 4 and Fig. S4).
In terms of the risk of hydraulic failure, out of the 1000 ensemble members, ~40% of the
simulations for February and ~60% of simulation for August suggest that branches are the most
vulnerable plant organ, based on highest loss of conductivity across the continuum from root
to branch (Fig. 6). For the dry month of February, roots are at greater risk in comparison to the
wet season. If we consider the loss of conductivity more than 50% for February 2016 as a
threshold for a high risk of mortality (Adams et al., 2017), then 53% of ensemble simulations
reach this threshold. The key parameters affecting the risk of mortality, as measured by
percentage difference in parameter values for ensemble members reaching 50% loss of
conductivity or not, include the water potential leading to 50% loss of conductance for stomata
(*p50_gs*), stem (*p50_node_stem*), and transporting roots (*p50_node_troot*), maximum
hydraulic conductivity of stem (*kmax_node_stem*), and the taper exponent (*p_taper*) (Fig. 7).
Ensemble members with high risk of mortality generally have a higher *p_taper* and
*kmax_node_stem*, less negative *p50_gs*, and less negative p50 for stem and transporting roots
(Fig. 8).
**4. Discussion**
Our analysis showed the importance of key plant hydraulic traits in simulating plant water
potential and risk of hydraulic failure. This analysis identifies these parameters as potential
targets of either model calibration or targeted measurement campaigns to achieve realistic
simulations. In our sensitivity analysis, the most influential parameter for both water potential
and loss of conductivity is the tapering of the radius of conduit with increasing plant height
(*p_taper*). As *p_taper* increases, the conduit radius increases from the top of the tree to its
base. According to Hagen-Poiseuille's equation, this increases the theoretical maximum total
conductance. Low values of *p_taper* thus limit the adverse effects of tree height by increasing
*k_max* along the whole continuum and reducing the soil-to-leaf water potential needed to
maintain transpiration. Our inference is that *p_taper* represents an overarching property of
plant architecture that influences the relative effect of each of the other traits related to
hydraulic safety and efficiency (Olson et al., 2021). The xylem architecture as determined by
*p_taper* parameter could change in response to age and development stages (Rodriguez-
Zaccaro et al., 2019), which is not considered in this study. Future studies evaluating the
importance of this change to hydraulic functions could be useful to guide size-dependent
growth and mortality. Another dimension of the hydraulic architecture with a critical role in
determining both water potential and loss of conductivity, though to a much lesser degree, was
the fraction of total tree resistance that is belowground (i.e., of the entire transporting and
absorbing root system; 1- *rfrac_stem*). Generally, a plant will match the growth of its trunk
and crown to maintain a degree of equilibrium in aboveground resistance as the distance water
needs to travel increases (Yang and Tyree, 1993). In this study, due to the lack of data on the
belowground resistance, we assigned a quite large range for this trait, which could be impacted
by many factors such as belowground root biomass, root network architecture, and interactions
between roots, fungi and bacteria (Poudel et al., 2021; Bhagat et al., 2021).
The second most sensitive parameter in determining loss of conductance was the leaf water
potential at 50% loss of stomatal conductance (*p50_gs*). This parameter controls the water loss
rate from leaves, with a less negative value providing protection from hydraulic failure during
water-limited periods. The *p50_gs* trait has been shown to play a key role in tree survival
during severe droughts (Breshears et al., 2009; Rowland et al., 2015). The ability to withstand
lower leaf water potentials is also a key indicator of sapling and seedling survival during
drought and determines species distribution across a moisture gradient (Kursar et al., 2009).
There may be a trade-off between drought tolerance (with a lower *p50_gs*) and drought
avoidance (a less negative *p50_gs* but with a high capacitance; the amount of water released
from reserves as leaf water potential declines), a crucial aspect in determining species drought
resistance (Pineda-Garcia et al., 2013). Additionally, loss of conductivity was sensitive to the
water potential at 50% loss of max conductivity within the stem (*p50_stem*) as it can largely
affect the whole plant conductance and thus the water supply to the leaves. *p50_stem*
negatively correlates with wood density and may be a marker of the trade-off between
hydraulic efficiency and safety within the stem (Chen et al., 2009; Manzoni et al., 2013);
however, other studies have shown that this trade-off is weak (Gleason et al., 2016). Liang et
al (2019) showed that the strength of this trade-off could be dependent on specie's drought
strategies.
Leaf water potential and loss of conductance were both sensitive to the maximum xylem
conductivity in the stem (*kmax_node_stem*). Higher maximum conductivity represents greater
xylem efficiency, which in the absence of drought or light limitations would result in greater
potential photosynthesis and less negative water potentials (Gleason et al., 2016). However,
xylem with higher *kmax_node_stem* could be more vulnerable to embolism as water potential
declines (Sperry and Love, 2015). In tropical rainforests, species with higher conductivity per
unit leaf area generally are less desiccation-tolerant, and thus exhibit higher mortality rates
(Kursar et al., 2009). Low *kmax_node_stem* along with high leaf-to-sapwood area ratio (*la2sa*)
also represents a vulnerability to reduced conductance which increases with height
(Christoffersen et al., 2016).

Traits with lower order of impacts on water potential modulate the amount of stored water

available during drought. The bulk modulus of elasticity in the root (*epsil_node_aroot*)
together with root saturated water content determines the amount of water available from
cellular storage between complete hydration and loss of turgor (Powell et al., 2017). This
represents the ability of the roots to continually supply water to the rest of the plant as drought
occurs. It also represents an investment in cellular structure, which may be an additional
indicator of adaptations with non-hydraulic origin. The residual water content in the stem
(*resid_node_stem*) determines the minimum amount of water xylem will hold and thus impact
the amount of water storage plant can use during drought as well (Bartlett et al. 2012). In this
study, we made the assumption that the traits are independent of each other, in order to
understand the hydrodynamic behaviors of FATES-HYDRO for different hydraulic traits
based on a single PFT. Understanding the trade-offs between these traits is crucial for
determining the competition among different PFTs. Future studies would greatly benefit from
assessing the significance of these trade-offs to predict vegetation dynamics under future
climate change.

In contrast to the majority of hydraulic traits in the model, conduit taper, the fraction of

total resistance belowground, and the leaf to sapwood area ratio are whole-plant hydraulic
traits. Our analysis highlights the importance of whole-plant hydraulic traits such as conduit
taper relative to tissue-level hydraulic traits for a range of plant hydraulic functions, including
whole-plant conductance and hydraulic failure risks. An important area for future work is to
better constrain and understand the consequences of intra- and interspecific variation in these
whole-plant hydraulic traits in tropical forests. Our choice of the range of variation in the
conduit taper exponent came from a study on temperate species, and was broad, encompassing
the entire range of observed values in that study (Savage et al. 2010). Further, we estimated
the effects of variation in the taper exponent on whole-plant conductance conditional on trees
following a simple set of optimality assumptions (space-filling, area-conserving, and self-
similar branching network structure). However, in practice, such assumptions are often not met
(Smith et al., 2014). Therefore, it is possible that the model sensitivity to xylem taper in terms
of whole-plant hydraulic function are overestimated. Nevertheless, our study highlights the
importance of better constraining this parameter as well as further experimentation with
alternate model structures to better account for non-optimal trees in tropical forests.
The sensitivity of vegetation to drought stress and hydraulic-failure-induced mortality is of
paramount importance for understanding how ecosystems may respond to shifting temperature
and rainfall patterns under a changing climate (Mcdowell et al., 2022). We recognize that
parametric sensitivity could be different for different sites depending on climate driver, soil
moisture and vegetation types. However, we expect the main parameter of importance could
be useful to guide model calibration to select the candidate parameters for different sites. As
understanding of plant hydrodynamics increases, linking model predictions to observable plant
traits has emerged as a promising means of constraining predictions of ecosystem resilience.
Such traits are challenging and costly to measure in the field and thus resources must be
directed carefully when planning measurement campaigns. The identified parameters in this
study could provide guidance on the limited measurement we could target in the field.
**5. Acknowledgment**
This research was supported as part of the Next Generation Ecosystem Experiments-
Tropics, funded by the U.S. Department of Energy, Office of Science, Office of Biological and
Environmental Research. RF acknowledges funding by the European Union's Horizon 2020
(H2020) research and innovation program under Grant Agreement No. 101003536 (ESM2025
– Earth System Models for the Future) and 821003 (4C, Climate-Carbon Interactions in the
Coming Century).
**6. CODE and Data Availability**
The FATES-HYDRO code is available from https://doi.org/10.5281/zenodo.7686333. The
traits data are in the supplementary file [traits_master_trop.csv].
**7. Supplement Information**
Three supplementary file are included. The HYDRO_DESCRIPTION.pdf provide the
summary of the hydrodynamic implementation that is different from Christoffersen et al.
(2016). The traits_master_trop.csv file include all the hydraulic traits we assembled for the
tropical region. The supplementary_figure.pdf provides additional figures for the main text.
**8. Author contribution**
CX and BC designed the sensitivity analysis experiments. BC collected the data and fitted
the trait distributions. CX conducted the analysis and drafted the manuscript. BC, CX, RF, RN
and CK designed the implementation of HYDRO codes. BC implemented the majority of
HYDRO codes with code improvement made by CX and RN. ZR conducted the ensemble
model simulations. MS provided the leaf cuticular conductance data. NM, CK and LK
provided guidance on the sensitivity analysis, code development and trait data synthesis. All
authors contributed to manuscript writing by providing edits and suggestions.
**9. Competing interests**
The contact author has declared that none of the authors has any competing interests.


**Figures**


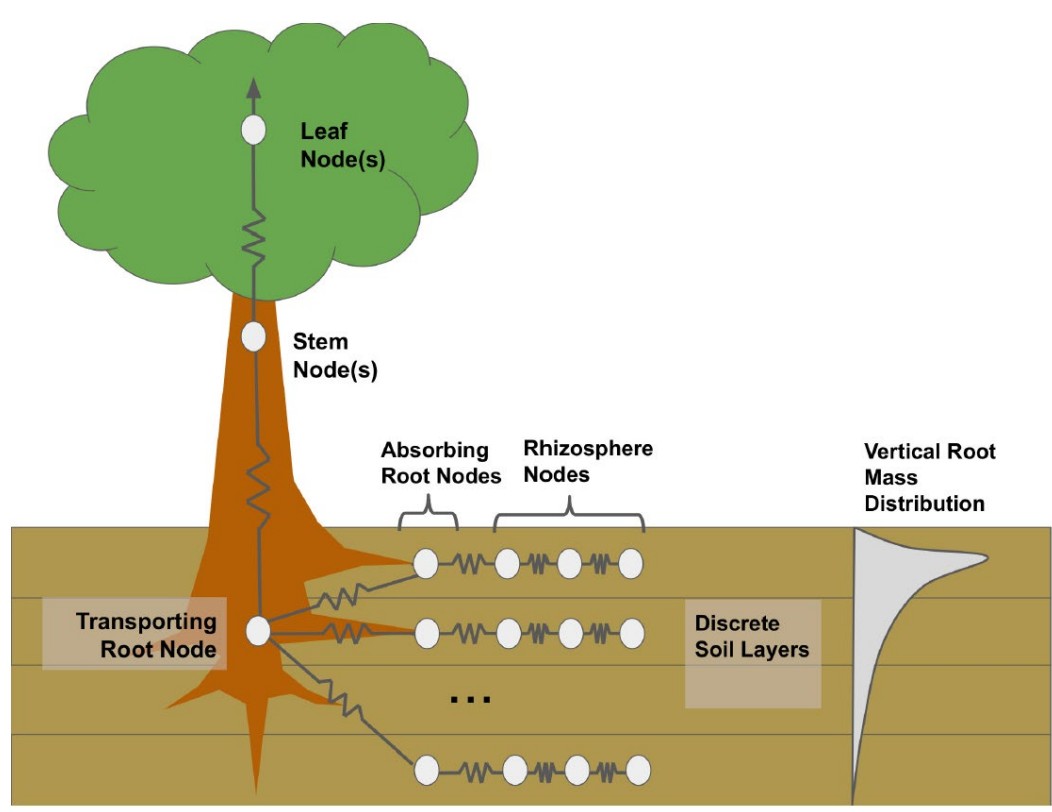


**Figure 1: Diagram of FATES-HYDO with simulation of rhizosphere shell, absorbing roots, transporting roots, stem and**
**leaves.** The model is solved for different soil layers with different root distributions.



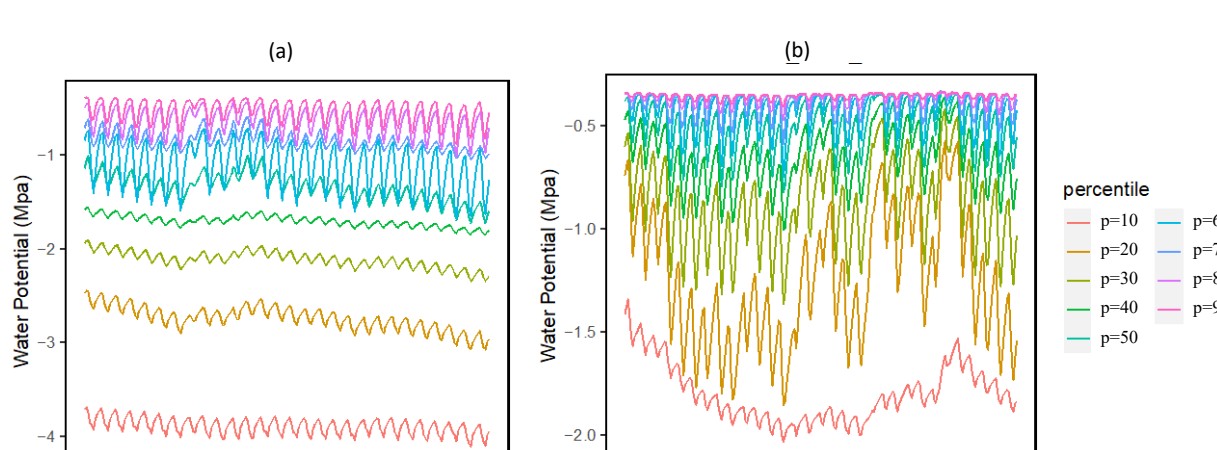


**Figure 2: Simulated ranges of leaf water potential for February (a) and August (a), 2016 for trees with DBH > 60cm.** The percentiles are calculated based on the monthly mean values of leaf water potentials for the 1000 ensemble simulations.



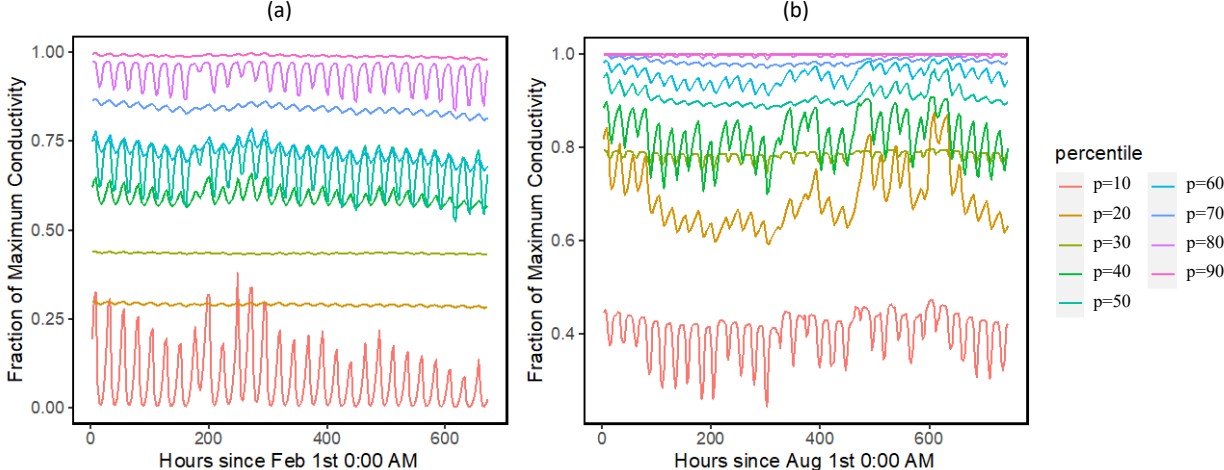


**Figure 3: Simulated ranges of fraction of maximum hydraulic conductivity of stem for February (a) and August (a), 2016**
**for trees with DBH > 60cm**. The percentiles are calculated based on the monthly mean values of leaf water potentials for the
1000 ensemble simulations.


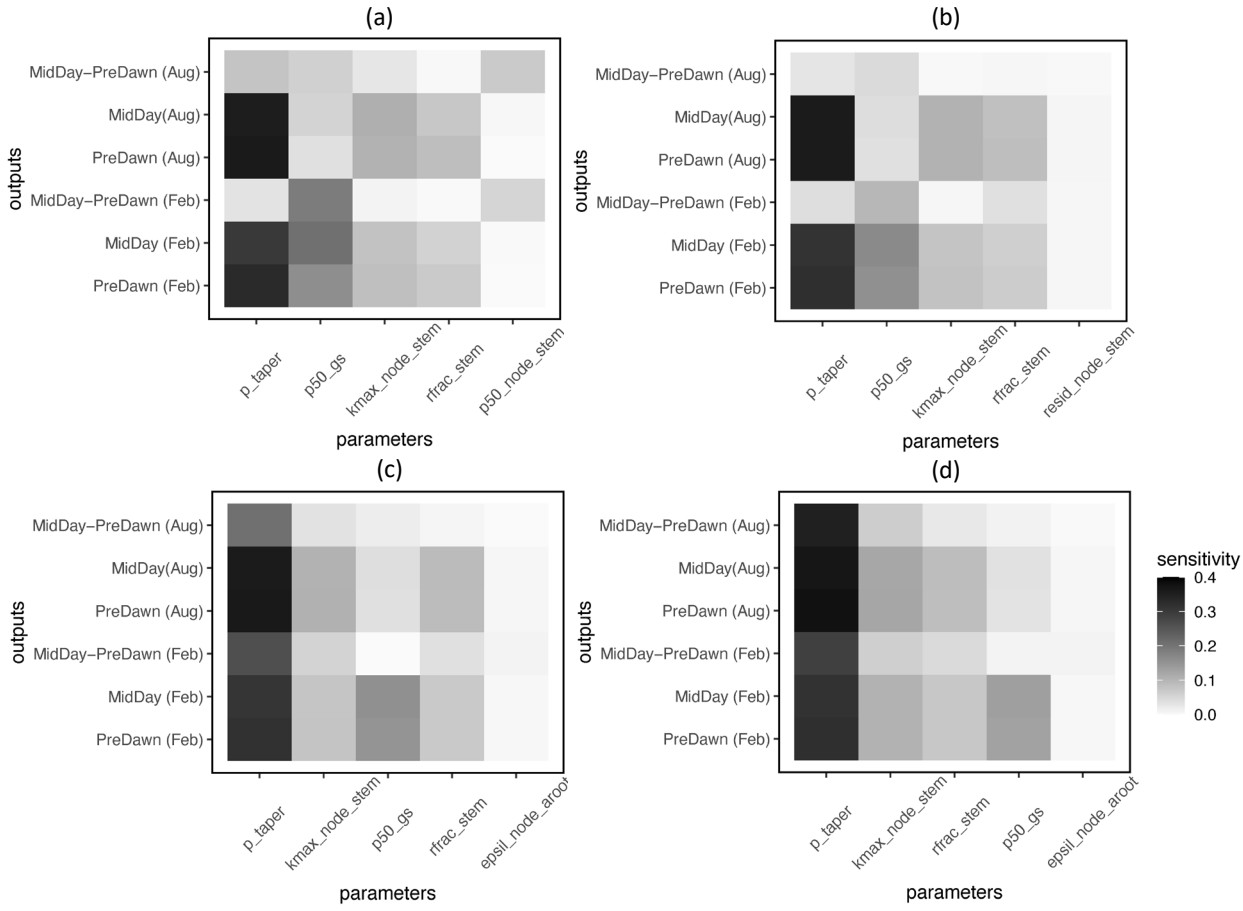

**Figure 4: Key parameters that control simulated water potentials for leaf (a), stem (b), transporting root (c) and absorbing root (d), for trees with DBH > 60cm.** The sensitivity value refers to the proportion of total model output variance contributed by a specific parameter (0-1). See Table 1 for the explanation of the parameters.

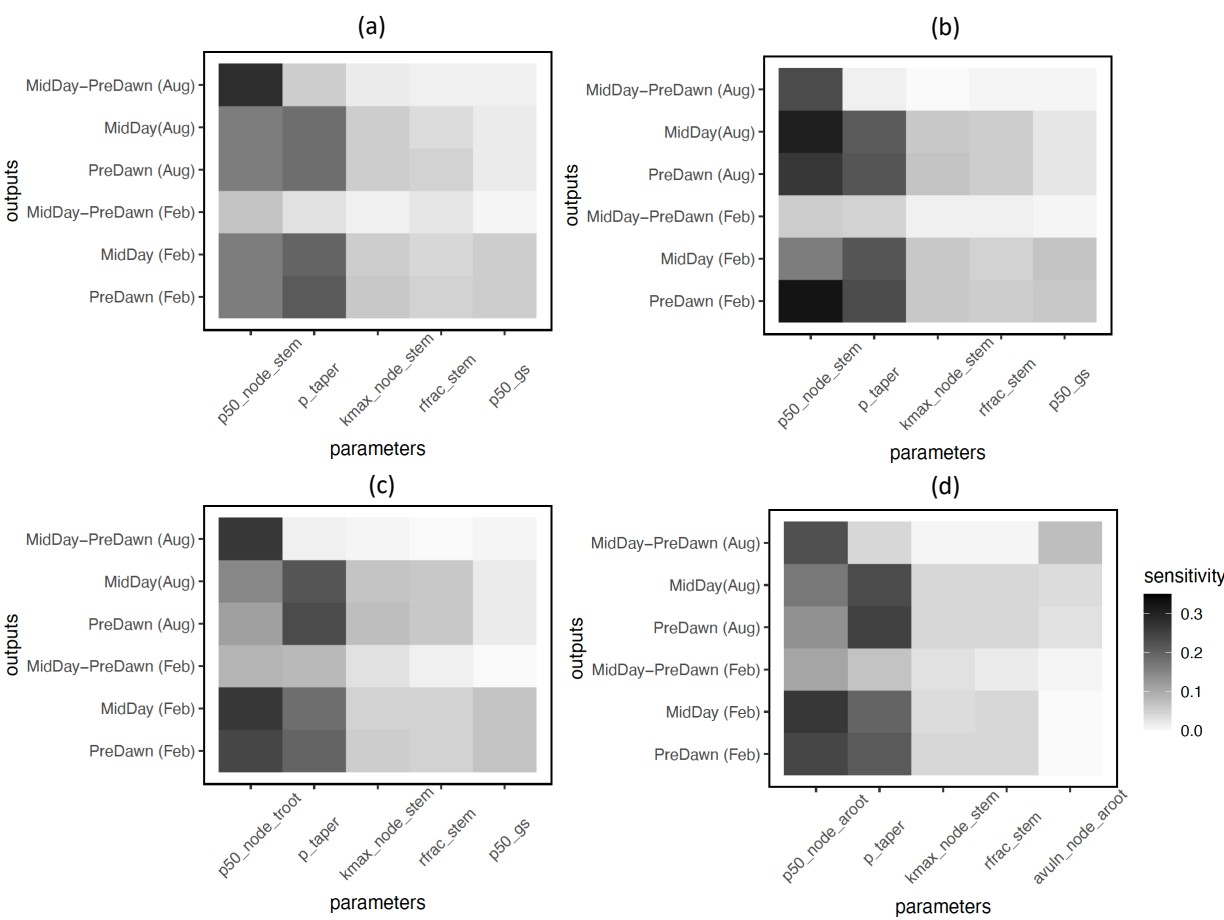

**Figure 5: Key parameters that control simulated loss of conductivity for branch (a), stem (b), transporting root (c) and absorbing root (d), for trees with DBH > 60cm.** The sensitivity value refers to the proportion of total model output variance contributed by a specific parameter. See Table 1 for the explanation of the parameters. See Table 1 for the description of parameters.

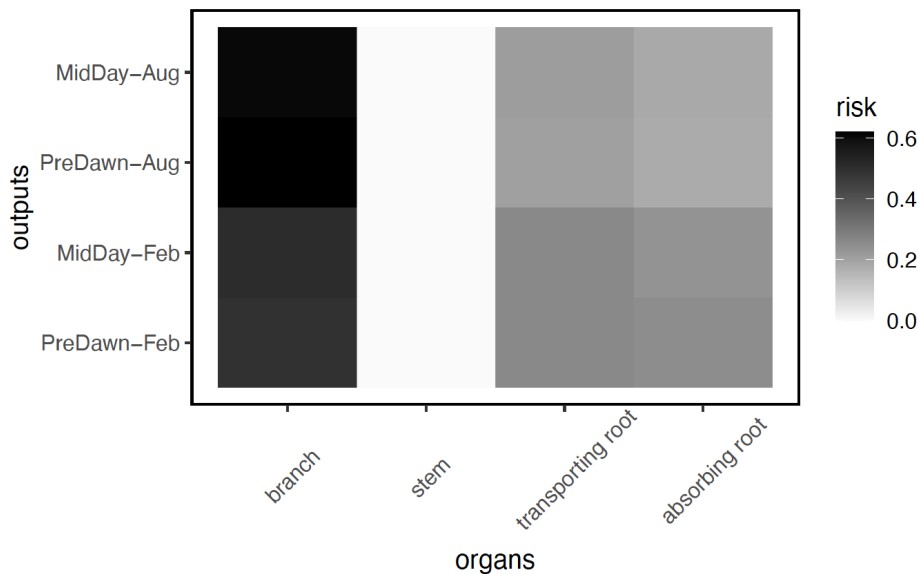

**Figure 6: Risk on the continuum for hydraulic failure as measured by percentage of total number of simulations with highest loss of conductivity for a specific organ (branch, stem, transporting root and absorbing root), for trees with DBH > 60cm.** As the model does not specifically simulate the branch, we calculated the risk of loss of conductivity based on the leaf water potential and hydraulic vulnerability curve from xylem.




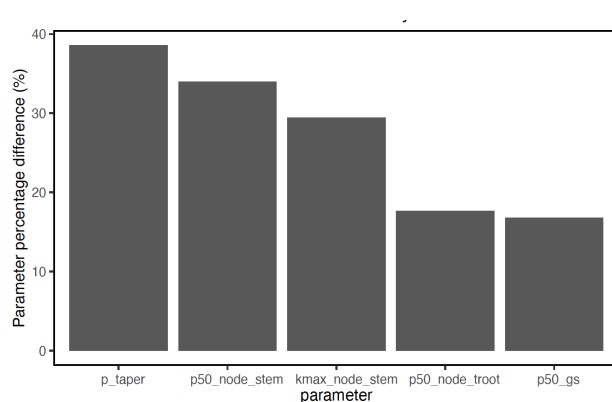


**Figure 7: Mean trait percentage difference for model ensemble simulations with loss of hydraulic conductivity larger than 50% and ensemble simulations with loss of hydraulic conductivity less than 50%, for trees with DBH > 60cm.** See Table 1 for the description of parameters.



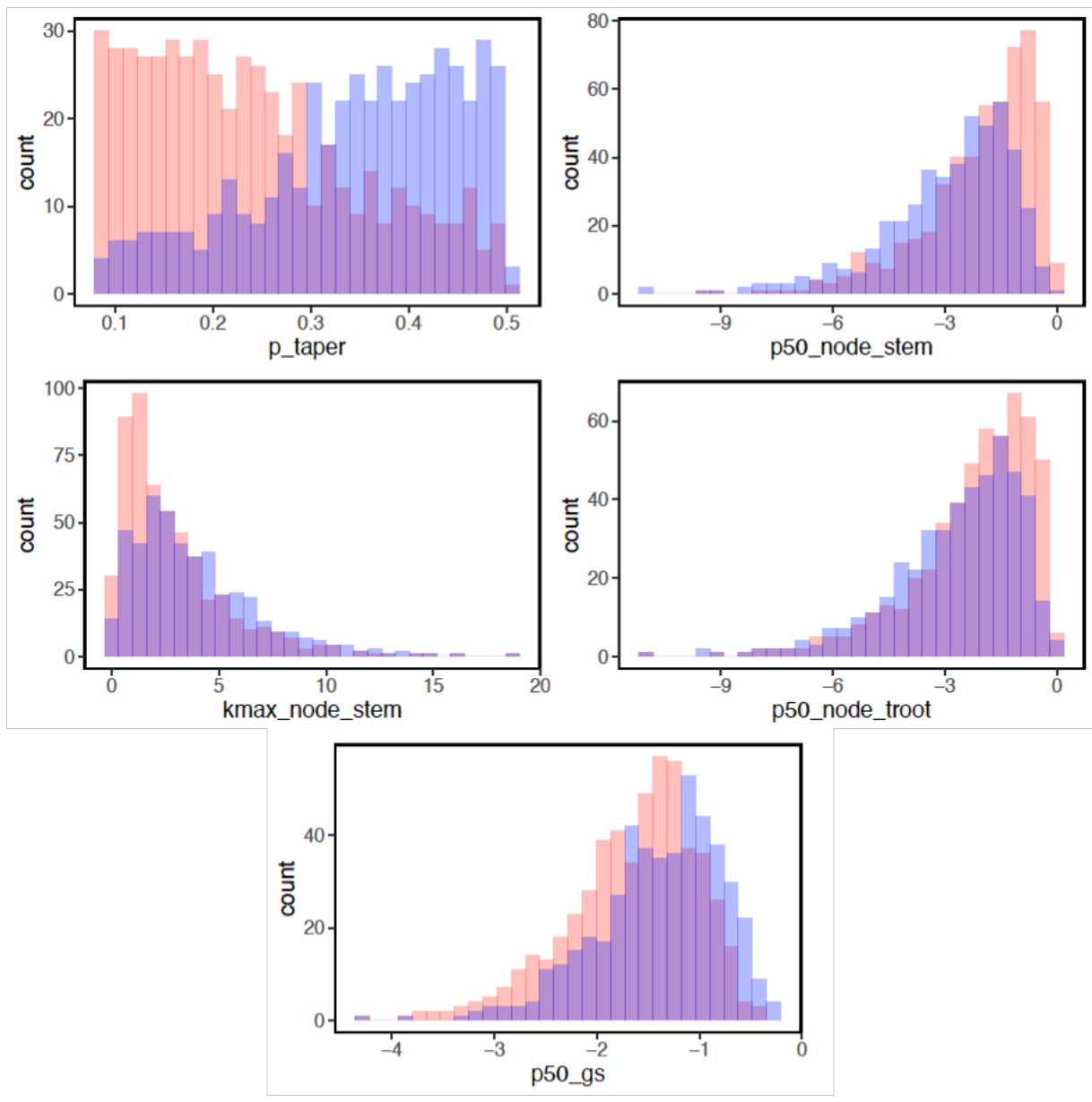


**Figure 8: Parameter difference for ensemble members with risk of mortality, for trees with DBH > 60cm.** Blue bars
indicate parameter values with lower mortality risk (<50% loss of hydraulic conductivity). Red bars indicate parameter values
with higher mortality risk (>= 50% loss of hydraulic conductivity) and purple bars indicate parameter values stacked from
transparent red/blue bars. See Table 1 for the description of parameters.



Table 1 Hydraulic parameters considered in the sensitivity analysis

| PARAMETER (EQUATION NUMBER)[1] | SYMBOL[2] | UNITS | DISTRIBUTION[3] | SOURCES& NOTES |
|---|---|---|---|---|
| **Pressure-Volume (PV) curve (water content – water potential relationship)** | | | | |
| saturated water content (thetas_node_leaf, thetas_node_stem, thetas_node_troot, thetas_node_aroot) (Eq. 5) | $\theta_{sat,x}$ | cm$^3$ cm$^{-3}$ | Leaf: Beta (9.69, 6.20) Stem: Beta (12.67, 7.4626) TRoot and ARoot: Beta (22.98, 5.29) | Christoffersen et al. (2016) Iversen et al. (2017) Wright et al. (2010) Roderick et al. (1999) Sack et al. (2003) Binks et al. (2016) |
| turgor loss point (tlp_node_leaf, tlp_node_stem, tlp_node_troot, tlp_node_aroot) (Eq. 5) | $\pi_{tlp,x}$ | MPa | $\pi_{tlp} = (\pi_0\,\varepsilon)/(\pi_0 + \varepsilon)$ | Bartlett et al. (2012) Christoffersen et al. (2016) |
| osmotic potential at full turgor (pinot_node_leaf, pinot_node_stem, pinot_node_troot, pinot_node_aroot) (Eq. 6) | $\pi_{0,x}$ | MPa | Leaf: G [9.8,6.26], Stem, TRoot, ARoot: LN [0.32,0.39] | Bartlett et al. (2012, 2014, 2016) Christoffersen et al. (2016) |
| bulk elastic modulus (epsil_node_leaf, epsil_node_stem, epsil_node_troot, epsil_node_aroot) (Eq. 7) | $\varepsilon_x$ | MPa | Leaf: G (4.07, 4.12) Stem, TRoot and ARoot: G [3.57, 3.84] | Bartlett et al. (2012, 2014) Christoffersen et al. (2016) |
| residual water fraction (resid_node_leaf, resid_node_stem, resid_node_troot, resid_node_aroot) (Eq. 5) | $RWC_{r,x}$ | unitless | Leaf: B [2.14,4.10] Stem, TRoot and ARoot: B [2.71, 4.53] | Bartlett et al. (2012, 2014) Christoffersen et al. (2016) |
| **Vulnerability Curve (water potential – hydraulic conductivity relationship)** | | | | |
| water potential at 50% loss of max conductivity (p50_node_stem, p50_node_troot, p50_node_aroot) (Eq. 3) | $P_{50,x}$ | MPa | Stem, TRoot and ARoot: G [2.07, 1.18] | Choat et al. (2012) |
| vulnerability curve shape parameter (avuln_node_stem, avuln_troot, avuln_node_aroot) (Eq. 3) | $a_x$ | unitless | Stem, TRoot and ARoot: LN [0.82,0.66] | Choat et al. (2012) |
| xylem conductivity per unit sapwood area (kmax_node_stem) (Eq. 8) | $k_{s,max}$ | kg m$^{-1}$ s$^{-1}$ MPa$^{-1}$ | G [1.41, 2.37] | Choat et al. (2012) |

| Leaf hydraulics | | | | |
|---|---|---|---|---|
| leaf water potential at 50% loss of max gs (p50_gs) (Eq. 12) | $P_{50,gs}$ | MPa | G [5.73, 0.27] | Klein (2014) |
| stomatal vulnerability shape parameter(avuln_gs) (Eq. 12) | $a_{gs}$ | unitless | $a_{gs}$= -2.406 P50,gs (-P50,gs) $^{-1.25}$ | Christoffersen et al. (2016) |
| Leaf cuticular conductivity (k0_leaf) (Eq. 11) | $g_0$ | umol m$^{-2}$s$^{-1}$ | LN [1.04, 0.84] | Slot et al. (2021) |
| **Plant Hydraulic Architecture** | | | | |
| Xylem taper exponent for sapwood (p_taper) (Eq. 9) | $p$ | (-) | U (0.08, 0.5) | Savage et al. (2010) |
| Leaf area to sapwood area ratio (la2sa) (Eq. 8) | $\dfrac{A_l}{A_s}$ | (-) | LN (-0.48, 0.77) | Choat et al. (2012) |
| **Root hydraulic Traits** | | | | |
| specific root length (*srl*) (Eq. 13) | $srl$ | m g$^{-1}$ | G [1.70, 35.31] | Iversen et al. (2017) |
| absorbing root radius (*rs2*) (Eq. 13) | $r$ | mm | LN [-1.91, 0.79] | Iversen et al. (2017) |
| fraction of total tree resistance that is aboveground (*rfrac_stem*) (Eq. 10) | $R_{frac,stem}$ | Unitless | U [0.1,0.7] | This study; empirical |
| root-soil interface conductivity per unit surface area ($K_{r1}$) (Eq. 14) | $k_{r1,max}$ | kg m$^{-1}$ s$^{-1}$ MPa$^{-1}$ | G [1.41, 2.37] | This study; empirically set the same as xylem conductivity |
| maximum root water loss rate ($K_{r2}$) (Eq. 14) | $k_{r2,max}$ | kg m$^{-1}$ s$^{-1}$ MPa$^{-1}$ | LN [-6.80, 0.92] | Wolfe (2020); Empirically set as 1/1000 bark water loss rate |

**Note:** 1: Several hydraulic parameters are used for different nodes of the plant including leaf, stem,
transporting root (troot), and absorbing root (aroot). For better reference in the text, we provided a list of
these parameters for specific nodes in the parenthesis; 2: Subscript *x* represents different tissue nodes in
the model;  3:B-Beta distribution;  U- Uniform distribution [lower limit, upper limit]; N-Gaussian
distribution (mean, standard deviation); LN-Log Normal Distribution [mean, standard deviation]; G-
Gamma distribution (lambda, scale); TRoot-Transporting root; ARoot-Absorbing root.

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
