# Peer review of "Quantification of hydraulic trait control on plant hydrodynamics and risk of hydraulic"

_EGUsphere, 2023_

## Author Comment (AC1)

**Response to Review #1**

**Comment:** This study tested the sensitivity of the parameters of a plant hydraulics model coupled in a demographic vegetation model (FATES-Hydro). This is an important step for model development and also helpful for understanding model behavior. The simulation experiments and analysis are solid, and the paper is generally well written. However, some places are not clear to me. I expect the authors can improve the description of the design of simulation experiments and the analysis of the results.

**Response**: Thank you for your time to review our manuscript and we will address your comments in the revised manuscript as detailed below.

Major questions/suggestions:

**Comment 1**. For the key parameters in Table 1, is it possible to list the key equations of this model that are related to these parameters? An analytical analysis of these equations would help to understand the sensitivity of these parameters.

**Response:** In the revised manuscript, we will add specific equations related to parameters in Table 1 and add these key equations to section 2.1 as well.

**Comment 2**. If I understand it correctly, the plant traits data of different species in the tropical forest of BCI are used to define the parameter ranges and distribution, from which the ensembles are sampled. This means a mean PFT is defined in each parameter combination. However, there are trade-offs in these parameters. How is this considered in the design of ensembles?

**Response**: Because our goal is to understand the model behaviors as determined by different hydraulic traits, we assumed independence among traits. Thus, we did not consider the tradeoff between traits. We will add caveats on the importance of trade off among hydraulic traits in discussion for future studies focusing on the uncertainty of model outputs.

**Comment 3**. The authors set the vegetation static (no growth, no reproduction and mortality). Please list the details of vegetation. Such as how many cohorts? what are the sizes of these cohorts?, etc.

If there was only one cohort in these tests, what is the size? Does the parameter sensitivity change with tree size? For example, the most important parameter according to this test, taper factor, may relate to the tree sizes.

**Response**: Thanks for these good points. The forest has 137 cohorts with diameters ranging from 10 cm to >2 meters. See the figure below for the size distributions. In the model output, we aggregated cohort information into different tree size class bins . In this

paper, we focused on the outputs for the size >60 cm bin, in view that they are most sensitive to canopy environmental conditions. Per your suggestion, we will add the sensitivity analysis to the size below 60cm in the revised manuscript.

[Figure]

**Comment 4**. In the discussion, some claims and opinions can be evidenced by recent research. Please add those references.

**Response**: We have added references in the discussion per our response to the detailed comments below.

Details comments:

**Comment**: Lines 36~38: about the statistical distribution of plant traits, please also clarify that they are used in parameter sampling (if they are).

**Response**: Yes, these traits are used in the parameter sampling. We will revise it as "... determined the best-fit statistical distribution for each trait, which is used in model parameter sampling to assess the parametric sensitivity".

**Comment:** Line 101: "we describe the implementation of a hydrodynamic scheme within FATES,": to me, this paper only tested parameter sensitivity, did not "describe the implementation of a hydrodynamic scheme". Am I wrong? If yes, please provide a detailed description of the hydro model.

**Response**: This manuscript is mostly focused on sensitivity analysis. The hydro code is based on Christoffersen et al.,( 2016), but with modifications made to be implemented in FATES. We did provide detailed implementation of the codes in FATES in the supplementary file [MODEL DESCRIPTION: updates made to TFS-HYDRO for FATES-HYDRO].

**Comment**: Line 102: "assess the importance of different hydraulic traits". I think it is about sensitivity. if it is "importance", then an index should be defined.

**Response:** We agree with your opinion here and will change 'importance' to 'sensitivity'.

**Comment**: Lines 115~116: "FATES simulates growth by integrating photosynthesis across different leaf layers for each cohort." From somewhere else, I learned that FATES does not have multiple leaf layers in a crown. Otherwise PPA principles cannot be applied. Please clarify.

**Response**: The PPA is applied based on the crown area to determine whether cohorts are positioned in the canopy and understory. For trees in both the canopy and understorey, there are indeed different layers of leaves. We hope this clarifies the confusion and we will add this to the revised manuscript.

**Comment**: Line 151: "we used the static stand structure mode of FATES": Please provide a detailed description of the static cohorts and tree sizes.

**Response**: We will add the number of cohorts and the size distribution of trees in the revised paper.

**Comment:** Lines 166~167: "here we focused on hydrodynamic behaviors for trees of diameter more than 60 cm.". Detailed cohorts and tree sizes please.

**Response:** We will add the number of cohorts and the size distribution of trees in the revised paper.

**Comment:** Line 169: "We identified 36 parameters for the FATES-HYDRO model (Table 1).". Please also provide the relevant equations.

**Response:** We will add a new column to add the relevant equations to the table. We will also add the relevant equations in the 2.1 and supplementary file.

**Comment**: Lines 181~194: I am not quite clear about this section. do authors use the multiple trees' traits to define a "mean state" tree?

**Response:** Each model ensemble member consists of a community of trees with identical traits (one PFT), but does not necessarily represent a "mean state" tree because we resampled from the observed distribution of trait values. We will reword the lines 179-184 for clarity as follows: "This trait dataset consisted of anywhere from 1 - 323 observations for each trait, where each observation corresponds to a different species (multiple observations for the same species are first averaged; see above). Before fitting distributions to these data, some traits were first transformed to be positive (e.g., P50) or normalized within [0, 1] when upper and lower bounds were well-defined (Table 1). Then, for each trait separately, we used the fitdistr package in R to estimate best-fit parameters for uniform, beta, normal, lognormal, and gamma statistical distributions in order to estimate central tendencies and spread for each trait. The distribution with the largest log likelihood and best-fit parameters are given in Table 1. Each model simulation consisted of a single PFT: all trees (across all cohort sizes and patches) had the same traits. The plant hydraulic traits in each simulation

were assigned using a random draw from each trait's distribution, and an ensemble of 1000 simulations were used to sample the observed plant hydraulic trait space for sensitivity analysis (see Section 2.3 Sensitivity analysis below)."

**Comment**: Lines 195~208, Section 2.3. I think an "important index" should be defined here. Or, make it clear that the most sensitive parameter is most important. (Though I don't think it is always true.)

**Response**: Agreed and we will add the following sentence to this section:

The parametric sensitivity index is calculated based on the ratio of the partial variance in the model output attributed to a specific parameter to the total variables in the model output.

**Comment**: Line 230 "during August compared to February". Please also note "wet' and "dry" season.

**Response**: We will add the wet/dry season in the parenthesis for each month in the revised manuscript.

**Comment:** Line 244: "1000 ensembles" should be defined in the method section. In the total 36 parameters, how many samples for each of them and how they combined?

**Response**: 1000 parameter values are sampled for each parameter and they are randomly combined. We will add this information to this section in the revised manuscript.

**Comment:** Lines 261~263: I guess p_taper is related to tree size. A test with different tree sizes can show if I am wrong.

**Response:** We will add the results of how p_taper will change with size in the revised manuscript.

**Comment:** Also p_taper comes with strong assumptions on plant development. Please cite some papers about that. There are some research on the changes in xylem structure with tree age.

**Response**: We will add the following sentence and citation to the revised manuscript:

The p_taper parameter determines the xylem architecture and it could change in response to age and development stages (Rodriguez-Zaccaro et al. 2019), which is not considered in this study. Future studies evaluating the importance of this change to hydraulic functions could be useful to guide size-dependent growth and mortality.

Rodriguez-Zaccaro, FD, Valdovinos-Ayala, J, Percolla, MI, Venturas, MD, Pratt, RB, Jacobsen, AL. Wood structure and function change with maturity: Age of the vascular cambium is associated with xylem changes in current-year growth. *Plant Cell Environ*. 2019; 42: 1816– 1831. https://doi.org/10.1111/pce.13528

**Comment:** Lines 270~271. Citation?

**Response:** We have rewritten it as follows:

While xylem taper exponent (p_taper), is a balance between maximizing conductance and hydraulic safety, it various through species and is additionally the product of maximizing carbon capture through leaf architecture and architectural and biochemical constraints (Savage et al., 2010).

Savage, V. M., Bentley, L. P., Enquist, B. J., Sperry, J. S., Smith, D. D., Reich, P. B., & Von Allmen, E. I. (2010). Hydraulic trade-offs and space filling enable better predictions of vascular structure and function in plants. Proceedings of the National Academy of Sciences, 107(52), 22722-22727.

**Comment:** Line 278 "interaction between root, fungi and bacteria.": citations are required here. I know there are some good review papers published in this area.

**Response**: We will add the following reference to the revised manuscript. We would appreciate any additional suggestions.

Bhagat, N., Raghav, M., Dubey, S. and Bedi, N., 2021. Bacterial exopolysaccharides: Insight into their role in plant abiotic stress tolerance,31(8): 1045-1059

---

## Author Response (AR1)

**Response to Review #1**

**Comment:** This study tested the sensitivity of the parameters of a plant hydraulics model coupled in a demographic vegetation model (FATES-Hydro). This is an important step for model development and also helpful for understanding model behavior. The simulation experiments and analysis are solid, and the paper is generally well written. However, some places are not clear to me. I expect the authors can improve the description of the design of simulation experiments and the analysis of the results.

**Response**: Thank you for your time to review our manuscript and we will address your comments in the revised manuscript as detailed below.

Major questions/suggestions:

**Comment 1**. For the key parameters in Table 1, is it possible to list the key equations of this model that are related to these parameters? An analytical analysis of these equations would help to understand the sensitivity of these parameters.

**Response:** In the revised manuscript, we have added specific equations related to parameters in Table 1 and added these key equations to section 2.1.

**Comment 2**. If I understand it correctly, the plant traits data of different species in the tropical forest of BCI are used to define the parameter ranges and distribution, from which the ensembles are sampled. This means a mean PFT is defined in each parameter combination. However, there are trade-offs in these parameters. How is this considered in the design of ensembles?

**Response**: Because our goal is to understand the model behaviors as determined by different hydraulic traits, we assumed independence among traits. Thus, we did not consider the tradeoff between traits. In the revised manuscript, we have added importance of trade off among hydraulic traits in discussion as follows:

"In this study, we made the assumption that the traits are independent of each other, in order to understand the hydrodynamic behaviors of FATES-HYDRO for different hydraulic traits based on a single PFT. Understanding the trade-offs between these traits is crucial for the competition among different PFTs. Future studies would greatly benefit from assessing the significance of these trade-offs to predict vegetation dynamics under future climate change."

**Comment 3**. The authors set the vegetation static (no growth, no reproduction and mortality). Please list the details of vegetation. Such as how many cohorts? what are the sizes of these cohorts?, etc.

If there was only one cohort in these tests, what is the size? Does the parameter sensitivity change with tree size? For example, the most important parameter according to this test, taper factor, may relate to the tree sizes.

**Response**: Thanks for these good points. The forest has 137 cohorts with diameters ranging from 10 cm to >2 meters. See the figure below for the size distributions. In the model output, we aggregated cohort information into different tree size class bins. In this paper, we focused on the outputs for the size >60 cm bin, in view that they are most sensitive to canopy environmental conditions. Per your suggestion, we will add the sensitivity analysis to the size below 60cm in the revised manuscript.

[Figure]

**Comment 4**. In the discussion, some claims and opinions can be evidenced by recent research. Please add those references.

**Response**: We have added references in the discussion per our response to the detailed comments below.

Detailed comments:

**Comment**: Lines 36~38: about the statistical distribution of plant traits, please also clarify that they are used in parameter sampling (if they are).

**Response**: Yes, these traits are used in the parameter sampling. We have revised it as "... determined the best-fit statistical distribution for each trait, which was used in model parameter sampling to assess the parametric sensitivity".

**Comment:** Line 101: "we describe the implementation of a hydrodynamic scheme within FATES,": to me, this paper only tested parameter sensitivity, did not "describe the implementation of a hydrodynamic scheme". Am I wrong? If yes, please provide a detailed description of the hydro model.

**Response**: This manuscript is mostly focused on sensitivity analysis but also introduce the HYDRO code. The HYDRYO code is based on Christoffersen et al.,( 2016), but with modifications made to be implemented in FATES.  We did provide detailed implementation of the codes in FATES in the supplementary file [MODEL DESCRIPTION: updates made to TFS-HYDRO for FATES-HYDRO].  We have now also provide more details on the

hydrodynamics in section 2.1 to lay out the key equations for a better understanding of the HYDRO code.

**Comment**: Line 102: "assess the importance of different hydraulic traits". I think it is about sensitivity. if it is "importance", then an index should be defined.

**Response:** We agree with your opinion here and we have changed 'importance' to 'parametric sensitivity'.

**Comment**: Lines 115~116: "FATES simulates growth by integrating photosynthesis across different leaf layers for each cohort." From somewhere else, I learned that FATES does not have multiple leaf layers in a crown. Otherwise PPA principles cannot be applied. Please clarify.

**Response**: The PPA is applied based on the crown area to determine whether cohorts are positioned in the canopy and understory. For trees in both the canopy and understorey, there are indeed different layers of leaves. We hope this clarifies the confusion and we have added this to the revised manuscript.

**Comment**: Line 151: "we used the static stand structure mode of FATES": Please provide a detailed description of the static cohorts and tree sizes.

**Response**: We have added the number of cohorts and the size distribution of trees in the revised paper and supplementary Fig. S1.

**Comment:** Lines 166~167: "here we focused on hydrodynamic behaviors for trees of diameter more than 60 cm.". Detailed cohorts and tree sizes please.

**Response:** We have added the number of cohorts and the size distribution of trees in the revised paper.

**Comment:** Line 169: "We identified 36 parameters for the FATES-HYDRO model (Table 1).". Please also provide the relevant equations.

**Response:** We have added the relevant equations to the table. We have also added the relevant equations in section 2.1.

**Comment**: Lines 181~194: I am not quite clear about this section. do authors use the multiple trees' traits to define a "mean state" tree?

**Response:** Each model ensemble member consists of a community of trees with identical traits (one PFT), but does not necessarily represent a "mean state" tree because we resampled from the observed distribution of trait values. We will reword the lines 179-184 for clarity as follows: "This trait dataset consisted of anywhere from 1 - 323 observations for each trait, where each observation corresponds to a different species (multiple observations for the same species are first averaged; see above). Before fitting distributions to these data, some traits were first transformed to be positive (e.g., P50) or normalized within [0, 1]

when upper and lower bounds were well-defined (Table 1). Then, for each trait separately, we used the fitdistr package in R to estimate best-fit parameters for uniform, beta, normal, lognormal, and gamma statistical distributions in order to estimate central tendencies and spread for each trait. The distribution with the largest log likelihood and best-fit parameters are given in Table 1. Each model simulation consisted of a single PFT: all trees (across all cohort sizes and patches) had the same traits. The plant hydraulic traits in each simulation were assigned using a random draw from each trait's distribution, and an ensemble of 1000 simulations were used to sample the observed plant hydraulic trait space for sensitivity analysis."

**Comment**: Lines 195~208, Section 2.3. I think an "important index" should be defined here. Or, make it clear that the most sensitive parameter is most important. (Though I don't think it is always true.)

**Response**: Agreed and we added the following sentence to this section:

… to estimate the parametric sensitivity index, which is calculated based on the ratio of the partial variance in the model output attributed to a specific parameter to the total variables in the model output.

**Comment**: Line 230 "during August compared to February". Please also note "wet' and "dry" season.

**Response**: We will add the wet/dry season in the parenthesis for each month in the revised manuscript.

**Comment:** Line 244: "1000 ensembles" should be defined in the method section. In the total 36 parameters, how many samples for each of them and how they combined?

**Response**: 1000 parameter values are sampled for each parameter and they are randomly combined. We added this information to this section in the revised manuscript as follows,

"For each ensemble simulation, each plant hydraulic trait was assigned with a random draw from each trait's distribution, and the samples for different traits are randomly combined to sample the observed plant hydraulic trait space for sensitivity analysis."

**Comment:** Lines 261~263: I guess p_taper is related to tree size. A test with different tree sizes can show if I am wrong.

**Response:** We added the results of the sensitivity analysis for trees less than 60cm in the results and supplemtary figures S3-4. We do observe that p_taper plays a less important role for smaller trees. We have added this result in the revised manuscript as follows:

"For smaller trees with diameter less than 60 cm, the corresponding parametric sensitivity patterns are similar to those of larger trees (Fig. S4 and Fig. S5); however, compared to larger trees, the parametric sensitivity of p_taper for simulated leaf water potential becomes lower for smaller trees (Fig. 4 and Fig. S4)."

**Comment:** Also p_taper comes with strong assumptions on plant development. Please cite some papers about that. There are some research on the changes in xylem structure with tree age.

**Response**: We have added the following sentence and citation to the revised manuscript:

The xylem architecture as determined by p_taper parameter could change in response to age and development stages (Rodriguez-Zaccaro et al. 2019), which is not considered in this study. Future studies evaluating the importance of this change to hydraulic functions could be useful to guide simulations of size-dependent growth and mortality.

Rodriguez-Zaccaro, FD, Valdovinos-Ayala, J, Percolla, MI, Venturas, MD, Pratt, RB, Jacobsen, AL. Wood structure and function change with maturity: Age of the vascular cambium is associated with xylem changes in current-year growth. *Plant Cell Environ*. 2019; 42: 1816– 1831.

**Comment:** Lines 270~271. Citation?

**Response:** We have rewritten it as follows:

"Our inference is that p_taper represents an overarching property of plant architecture that influences the relative effect of each of the other traits related to hydraulic safety and efficiency (Olson et al. 2021)."

Olson, M.E., Anfodillo, T., Gleason, S.M. and McCulloh, K.A., 2021. Tip-to-base xylem conduit widening as an adaptation: causes, consequences, and empirical priorities. New Phytologist, 229(4), pp.1877-1893.

**Comment:** Line 278 "interaction between root, fungi and bacteria.": citations are required here. I know there are some good review papers published in this area.

**Response**: We added the following references to the revised manuscript. We would appreciate any additional suggestions.

Bhagat, N., Raghav, M., Dubey, S. and Bedi, N., 2021. Bacterial exopolysaccharides: Insight into their role in plant abiotic stress tolerance, Journal of Microbiology and Biotechnology, 31(8): 1045-1059

Poudel, M., Mendes, R., Costa, L. A., Bueno, C. G., Meng, Y., Folimonova, S. Y., Garrett, K. A., and Martins, S. J.: The role of plant-associated bacteria, fungi, and viruses in drought stress mitigation, Frontiers in microbiology, 12, 3058, 2021.

**Response to Review #2**

**Comment**: The work by Xu et al. implemented a more trait-based model into FATES, and explored how the variation in traits may impact model simulations hence to test the models' sensitivity to those hydraulic traits. The manuscript is well written and well delivered. However, there are some major concerns over the manuscript given its positioning.

**Response:** Thank you for the time to review and we have revised the manuscript to address the concerns raised below.

**Comment 1**. It is not clear whether the manuscript is a model paper or validation paper. If the former, there were basically no details about the formulations; if the latter, the manuscript still lacks a fair amount of details for readers to understand how the traits are related to the modeling of vegetation processes. It seems that Lambert et al. (2022) GMD doi:10.5194/gmd-15-8809-2022 has more details on FATES-Hydro, but is not referenced in this study. I can see that the two studies have different aims, but this study should contain adequate details as Lambert et al. (2022).

**Response**: This manuscript mostly focused on the sensitivity analysis and the hydro code is based on Christoffersen et al.( 2016).  We did provide detailed implementation of the codes in FATES in the supplementary file [MODEL DESCRIPTION: updates made to TFS-HYDRO for FATES-HYDRO]. Per the suggestion from Review #1, in the revised manuscript, we have added key equations related to each parameter in the section of 2.1 and the supplementary file. We also added Lambert et al. (2022) GMD doi:10.5194/gmd-15-8809-2022  to our reference and the revised manuscript now provide as much detail as Lambert et al. (2022).

**Comment 2**. Following comment 1, these should be explicitly described in the manuscript:

> How canopy RT is done
> How canopy energy balance is done
> How the key parameters like taper component, Kmax, P50, Gs50, and etc are related to stomatal control
> How soil water balance is done, it is impacted by root distribution?
> What is the hydraulic architecture, number of roots, branches, and leaves, is there a trunk?
> How is sap area computed

Without these details, it is impossible to tell what is going on.

**Response**: Please see our response below specifically on different components. In the revised manuscript, we have provided a summary of each component and provide reference for the details to the reader of interest. Please see the details below:

- How canopy RT is done?

Canopy radiative transfer is calculated using a multi-layer scheme based on the iterative Norman radiation scheme. Leaf and stem area is binned into a matrix of canopy layer, leaf layer and plant functional types. Reflectance, abrogation and transmittance are calculated for each leaf layer. Between canopy layers, light streams are averaged between PFTs, such that all PFTs in the canopy layer below receive equal radiation on their top leaf layer. Fractional absorption of visible and near infra-red light is calculated separately for direct and diffuse light. For the direct stream, transmitted and reflected light is converted into diffuse fluxes. In FATES, the absorbed PAR is used to calculate photosynthesis rates for each of the canopy layer x leaf layer x PFT bins, after which rates across layers are re-aggregated into cohort level carbon fluxes. Please see the Supplementary file in Fisher et al. (2015) for details.

Fisher, R.A., Muszala, S., Verteinstein, M., Lawrence, P., Xu, C., McDowell, N.G., Knox, R.G., Koven, C., Holm, J., Rogers, B.M. and Spessa, A., 2015. Taking off the training wheels: the properties of a dynamic vegetation model without climate envelopes, CLM4. 5 (ED). *Geoscientific Model Development*, *8*(11), pp.3593-3619.

- How canopy energy balance is done

The energy balance is handled by the host land model. In this study, it is based on the land component of DOE's Exascale Energy Earth System Model (E3SM). The E3SM land model (ELM) is based on the Community Land Model 4.5 (Oleson et al 2013). Specifically, in ELM, the average canopy temperature is calculated based on the energy balance of latent heat, sensible heat, and absorbed radiation as determined by the radiative transfer model (above). The latent heat is determined by the transpiration, which is determined by the vapor pressure deficit from inside of leaf to the air, canopy stomatal conductance, and boundary layer constance. FATES calculated mean canopy stomatal conductance averaged across different cohorts, which is fed to ELM to calculate the energy balance. The Newton-Raphson numerical scheme is used to solve for the canopy temperature.

Oleson, K. W., Lawrence, D. M., Bonan, G. B., Drewniak, B., Huang, M., Koven, C. D., Levis, S., Li, F., Riley, W. J., Subin, Z. M., Swenson, S. C., Thornton, P. E., Bozbiyik, A., Fisher, R., Heald, C. L., Kluzek, E., Lamarque, J.-F., Lawrence, P. J., Leung, L. R., Lipscomb, W., Muszala, S., Ricciuto, D. M., Sacks, W., Sun, Y., Tang, J., & Yang, Z.-L. (2013). Technical description of version 4.5 of the Community Land Model (CLM) (*Tech. Rep. NCAR/TN-503+STR*). Boulder, Colorado, USA: National Center for Atmospheric Research.

- How the key parameters like taper component, Kmax, P50, Gs50, and etc are related to stomatal control

The means by which plant hydraulic traits and leaf water potential interact to influence stomatal control remains unchanged from Christoffersen et al. (2016), by replacing the default stomatal closure parameter with a prediction based on leaf water potential. At each 30-minute timestep, the model solves for updated water

potentials throughout the tree (leaf, stem, transporting and absorbing roots) based on the current timestep individual tree transpiration rate. The new leaf water potential is then used in the next time step to update a dimensionless stomatal closure parameter (beta; 0=fully closed;1=fully open), which impacts host land model (HLM) canopy energy balance in the standard way, and thus the simulated transpiration by the combined FATES canopy energy balance solution (see above).

- How soil water balance is done, it is impacted by root distribution?

All aspects of soil water balance (infiltration, water transfer among soil layers, and drainage) happen at the 'column' scale at 30-min timesteps and are handled within the Host Land Model (see Oleson et al. 2013 for a detailed description of hydrology in CLM4.5, the parent model of ELM, which is used in this manuscript). FATES-HYDRO handles soil water operations at the patch and cohort scales. It simulates root water uptake and changes in plant water potential from roots to leaves based on current timestep transpiration. The belowground conductance for each soil layer is weighted by root biomass with an exponential vertical distribution. Sections 2 and 3 in the Supplement of this manuscript provide full details on boundary conditions, sequence of operations among HYDRO and the HLM, downscaling of soil moisture to rhizosphere shells, and downscaling of transpiration from the patch to individual scale.

Oleson, K. W., Lawrence, D. M., Bonan, G. B., Drewniak, B., Huang, M., Koven, C. D., Levis, S., Li, F., Riley, W. J., Subin, Z. M., Swenson, S. C., Thornton, P. E., Bozbiyik, A., Fisher, R., Heald, C. L., Kluzek, E., Lamarque, J.-F., Lawrence, P. J., Leung, L. R., Lipscomb, W., Muszala, S., Ricciuto, D. M., Sacks, W., Sun, Y., Tang, J., & Yang, Z.-L. (2013). Technical description of version 4.5 of the Community Land Model (CLM) (*Tech. Rep. NCAR/TN-503+STR*). Boulder, Colorado, USA: National Center for Atmospheric Research.

- What is the hydraulic architecture, number of roots, branches, and leaves, is there a trunk?

The model is based on a beam approximation for each tree according to the Shinozaki pipe model (Shinozaki et al. 1964), in which the hydraulic path length from the trunk base to each leaf is assumed constant. A tree is approximated with single pools of water separately for each of leaves, stem (includes trunk and branches), transporting and absorbing roots with connecting resistors. This is shown in Figure S1 below. To better help the reader to better understand the structure, we have moved this S1 figure to the main text.

[Figure]

Shinozaki, K., Yoda, K., Hozumi, K. and Kira, T., 1964. A quantitative analysis of plant form-the pipe model theory: I. Basic analyses. *Japanese Journal of Ecology*, *14*(3), pp.97-105.

- How is sap area computed

The sapwood area is calculated based on the product of the leaf area and the ratio of leaf area to sapwood area, which is an input parameter in Table 1.

Minor comments:

**Comment:** Line 2: (FATES-HYDRO V1.0) or using FATE-HYDRO v1.0?

**Response**: it should be FATES-HYDRO V1.0 and it is revised in the manuscript.

**Comment:** Line 39: P50 for xylem or stomata? Need to be consistent, say P50x, P50gs

**Response**: We have made them consistent in the revised manuscript.

**Comment:** Line 41: top 5 traits? I can only found 4 from the text…

**Response**: Thanks for pointing this out. The sentence is updated in the revised manuscript as follows:

"We show that, for simulated leaf water potential and loss of hydraulic conductivity across different plant organs, the four most important traits were associated with xylem conduit taper (buffers increasing hydraulic resistance with tree height), stomatal sensitivity to leaf water potential, maximum stem hydraulic conductivity, and the partitioning of total hydraulic resistance above vs. belowground."

**Comment:** Line 86: such water limitation functions (based on soil moisture? to be more explicit)

**Response:** we have reworded it as 'soil-moisture-dependent water limitation functions' in the revised manuscript.

**Comment:** Lines 138-139: a function of the tissue water content? Why water content? Shouldn't it be xylem pressure?

**Comment:** Line 203: Sensitivyt or Sensitivity?

**Response:** It should be Sensitivity and it is revised in manuscript.

**Comment:** Line 245: branches are most vulnerable... How about leaves? Does this branch mean stem and leaf?

**Response**: Here, 'branch' includes to the tip of the leaf petiole; the model does not explicitly consider xylary or extraxylary resistance within and outside the leaf midrib. Thus, the vulnerability of leaf conductance is not explicitly simulated in the model. We have pointed this out in the revised manuscript.

**Comment:** Line 280: How is p50_gs used? Does it mean gs is always a function of Pleaf? Regardless of variations in PAR, CO2, VPD, and Psoil?

**Response**: The stomatal conductance (gs) is estimated based on the Ball-Berry model, with a slope (g1) and intercept (g0) to link g_s to humidity (RH), CO2 and photosynthetic rate (A) . A is determined by PAR and CO2 based on the Farquhar photosynthesis model. Namely,

$$gs = g0 + g1\ A\ RH/[CO2].$$

p50_gs is used to calculate a water limitation factor (Btran) based on leaf water potential, which is resulted from water loss from leaf and root water uptake as determined by soil water potential and plant as follows,

$$Btran = 1/\ (\ 1+ (Pleaf/p50\_gs)^a).$$

Btran is then applied to both gs and g0 to estimate its impact on gs. We have added this detail with equations to the revised manuscript to clarify this confusion.

**Comment:** Line 311: epsil_node, you need to be consistent with ths symbols (you provided two for the same parameter in Table 1)

**Response:** It is updated as 'epsil_node_root' in the revised manuscript.

**Comment:** Fig. 1 is too crowded, consider use fewer curves

**Response**: Agreed and in the revised manuscript, we have updated the plot for every 10th percentile.

**Comment:** Fig. 2 Xylem cavitation can fully recover?

**Response**: In this version of code, we assume that xylem cavitation can fully recover as long as the trees do not die. We have pointed this out in the revised manuscript after eq. (4).

---

## Author Response (AR2)

**Response:** Dear editor, thank you so much for the detailed check on our revision. Please see below our responses. We hope that our revised paper is ready for publication.

1. All figures with different lines for percentiles: Percentile values should range 10, 20, ..., 90 instead of 0.1, 0.2, ..., 0.9.

**Response:** We have now updated the figures in the main text (P29-30 in the revised manuscript) and the supplementary figure file (Page 3-4) as suggested.

2. This comment from Reviewer 2 was not addressed: "Lines 138-139: a function of the tissue water content? Why water content? Shouldn't it be xylem pressure?"

**Response:** Thank you for pointing this out. We have now updated it as water potential (Line 155, page 8 in the revised manuscript).

3. You mention addressing this comment from Reviewer 2, but I can't find where you did so: "Line 245: branches are most vulnerable... How about leaves? Does this branch mean stem and leaf?"

**Response:** We have added the following sentence to the revised manuscript (Lines 396-400, Page 19)**:** The HDYRO model only considers the stem node (Fig. 1) without explicitly simulating the branch. In this analysis, we calculated the branch vulnerability by using the PLC curve of xylem and the leaf water potential, which approximates the water potential at the tip of the branch. The model does not explicitly consider xylary or extraxylary resistance within and outside the leaf midrib.

4. L230: Please mention that both gs and g0 are multiplied by this stress factor.

**Response:** We have reworded the sentence as "Stomatal conductance (i.e., both $g_0$ and $g_1$) is further modified by a plant water stress factor…", in Line 230-231, Page 12 in the revised manuscript.

5. I'm confused about Reviewer 2's comment, "Line 311: epsil_node, you need to be consistent with ths symbols (you provided two for the same parameter in Table 1)". I only see one symbol for that parameter in Table 1. I'm also confused about your response, in which you change the name from epsil_node to epsil_node_root in the text but not the table.

**Response:** Several hydraulic parameters are used for different nodes of the plant including leaf, stem, transporting root (troot), and absorbing root (aroot). To avoid the confusion, we have now updated the table to list all the specific nodes for one hydraulic traits. For example, for saturated water content, we have listed thetas_node_leaf, thetas_node_stem, thetas_node_troot, thetas_node_aroot in the parenthesis and provided Note 1 below the table to clarify (Page 36 in the revised manuscript).

---

## Author Response (AR3)

**Response to editor**

**Comment:** Thanks for your revisions. I just noticed a typo at line 397 ("HDYRO"); when you submit a corrected version it'll go straight to the publication process.

**Response:** Thanks for catching this typo and we have corrected it in the final production file uploaded (Line 397).  Thanks for your great efforts to handle our manuscript.